# Advancements in Phase Change Materials in Asphalt Pavements for Mitigation of Urban Heat Island Effect: Bibliometric Analysis and Systematic Review

**DOI:** 10.3390/s23187741

**Published:** 2023-09-07

**Authors:** Claver Pinheiro, Salmon Landi, Orlando Lima, Larissa Ribas, Nathalia Hammes, Iran Rocha Segundo, Natália Cândido Homem, Verônica Castelo Branco, Elisabete Freitas, Manuel Filipe Costa, Joaquim Carneiro

**Affiliations:** 1Centre of Physics of Minho and Porto Universities (CF-UM-UP), Azurém Campus, University of Minho, 4800-058 Guimarães, Portugal; orlandojunior.jr@hotmail.com (O.L.J.); nathi_hammes@hotmail.com (N.H.); iran@fisica.uminho.pt (I.R.S.); carneiro@fisica.uminho.pt (J.C.); 2Federal Institute of Education, Science and Technology Goiano, Rio Verde 75901-970, Brazil; salmon.landi@ifgoiano.edu.br; 3University of Minho, ISISE, ARISE, Department of Civil Engineering, 4800-058 Guimarães, Portugal; 4Transportation Engineering Department, Federal University of Ceará, Fortaleza 60455-760, Brazil; larissa.ribas@det.ufc.br (L.R.); veronica@det.ufc.br (V.C.B.); 5Department of Transport Engineering and Geodesy, Federal University of Bahia, Salvador 40210-630, Brazil; 6Simoldes Plastics, 3720-502 Oliveira de Azeméis, Portugal; natalia.homem@outlook.com; 7Centre of Physics of Minho and Porto Universities (CF-UM-UP), Gualtar Campus, University of Minho, 4710-057 Braga, Portugal; mfcosta@fisica.uminho.pt

**Keywords:** cool pavements, enthalpy of fusion, leakage prevention, mechanical properties, melting point temperature, phase change materials (PCM), sustainable urban heat management, thermal performance, urban heat island (UHI) effects

## Abstract

This research presents a dual-pronged bibliometric and systematic review of the integration of phase change materials (PCM) in asphalt pavements to counteract the urban heat island (UHI) effect. The bibliometric approach discerns the evolution of PCM-inclusion asphalt research, highlighting a marked rise in the number of publications between 2019 and 2022. Notably, Chang’an University in China has emerged as a leading contributor. The systematic review addresses key questions like optimal PCM types for UHI effect mitigation, strategies for PCM leakage prevention in asphalt, and effects on mechanical properties. The findings identify polyethylene glycols (PEGs), especially PEG2000 and PEG4000, as prevailing PCM due to their wide phase-change temperature range and significant enthalpy during phase transitions. While including PCM can modify asphalt’s mechanical attributes, such mixtures typically stay within performance norms. This review emphasises the potential of PCM in urban heat management and the need for further research to achieve optimal thermal and mechanical balance.

## 1. Introduction

The intricate relationship between asphalt and the urban heat island (UHI) phenomenon is significant. UHIs arise where urban temperatures substantially exceed those of neighbouring rural areas [1]. A substantial contributor to this disparity is the prevalence of materials like asphalt in cityscapes. Its unique thermal properties enable it to absorb and retain vast amounts of solar radiation. Consequently, the surface temperature of asphalt can skyrocket under direct sunlight, primarily due to its dark hue and low thermal conductivity [2]. As day transitions to night, this stored heat is progressively released, further amplifying urban temperatures [3]. Other urban characteristics, such as towering buildings and a conspicuous absence of vegetation, obstruct airflow and impede the natural dissipation of heat [3]. Moreover, several other elements, like local climate, urban design, and construction density, intensify the UHI effect [3]. Recognising the severe implications of UHIs, which span from health hazards to heightened energy consumption and potential ecological ramifications, there is an urgent call for remedial measures [2].

Innovative interventions, such as introducing reflective ‘cool pavements’, aim to curtail heat absorption by reflecting a larger share of solar radiation [4]. Additionally, permeable pavements have emerged, facilitating water seepage and subsequent evaporation and cooling of the surroundings. An up-and-coming solution is the inclusion of phase change materials (PCM) within asphalt structures. By storing and releasing heat during their phase transitions, PCM can effectively modulate asphalt surface temperatures, mitigating the effects of urban heating [3]. It is fundamental to understand that while asphalt plays a key role in urban heating, its influence is intertwined with numerous other factors. Elements such as the local climate, the magnitude and density of urban development, the extent of vegetation cover, and the presence of water bodies collectively shape the urban thermal landscape [5]. Consequently, a holistic, multifaceted approach tailored to the nuances of each urban setting is imperative.

The transport infrastructure, especially road networks, undeniably reinforces a nation’s socioeconomic growth [1]. Within rapid urbanisation and industrial expansion, broadening transport routes for people and goods is necessary. This ensures that environmental repercussions, particularly those stemming from alterations to natural habitats, are minimised. Paved roads, primarily constructed using asphalt concrete, play a particular role in increasing the UHI phenomenon, wherein cities experience temperatures notably warmer than those of their rural counterparts [6,7] and urban microclimates are relatively more generous than surrounding rural environments. Carnielo and Zinzi have shown that the intensities of UHIs can reach 12 °C [8]. As a result of this phenomenon, increased demand for air conditioning can be highlighted. In developed countries, almost 8% of the total electricity generated is used to meet this exigency [9]. Furthermore, the reduction in human thermal comfort outdoors can inhibit the choice of walking and cycling even for short distances, which can aggravate various problems related to the increased flow of conventional vehicles, such as air pollution and risks to human health […].

Many studies have shown that the service life of asphalt pavements is substantially affected by high temperatures that occur during specific periods of the day. In addition, the vehicle load on traditional pavement, the surface temperature of which in summertime can reach 60 °C or more, provokes permanent deformations, ruts, and cracks, considerably reducing pavement safety [10]. It is known that when substances transform from the crystalline-solid phase to the amorphous-solid phase, solid to liquid, solid to gas, or liquid to gas, they absorb heat and vice versa. At the same time, there is no significant change in their temperature. In this context, PCM, which can regulate working temperatures by absorbing and releasing heat during changes in the physical state of matter, have been incorporated into asphalt concrete pavements to reduce temperature gradients. In this way, by avoiding excessive and abrupt changes in system temperatures, it is expected that both the deleterious effects of UHIs and the damage and maintenance costs of asphalt pavements can be reduced using PCM. For this purpose, PCM have been added to asphalt as an aggregate replacement before compaction (dry approach) or mixed with asphalt binder before adding the aggregate (wet method). As a third approach, acrylic paint with a PCM has been applied on the pavement surface [11]. Numerical simulations and experimental studies show that the addition of PCM positively affects the temperature regulation of asphalt pavements. Therefore, research in this field has intensified recently (Figure 1).

When asphalt mixtures are produced, the temperature can reach 180 °C. In this case, it becomes a challenge to synthetise PCM that do not suffer thermal decomposition when used to obtain cool pavements. Therefore, it is fundamental to investigate the thermal stability of PCM, which, in general, is performed using a thermogravimetric analyser. In addition, if the phase transition of the PCM in question is solid to liquid, it will probably be necessary to use a material that acts as a capsule or support for the PMC to prevent its leakage, which could even hurt the mechanical properties of the pavement [12,13]. In this case, the literature fundamentally presents two possibilities: (i) encapsulation (by melamine–formaldehyde resin [14], acrylic-based polymers [15], and CaCO_3_ [16], among others) or (ii) impregnation using porous materials (silica fume [17], diatomite [18], expanded graphite, and others [19]) or crystalline/lamellar materials (expanded graphite [20], reduced graphene oxide [21], and montmorillonite [22] (Figure 1).

**Figure 1 sensors-23-07741-f001:**
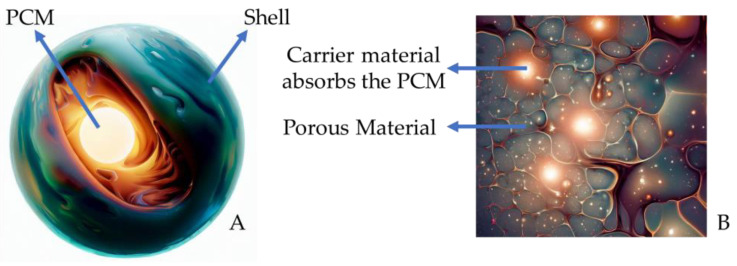
PCM presented as a core–shell system (**A**) and porous carrier material (**B**) (adapted) [11,23].

It is well known that thermal conductivity plays a key role in heat transfer. The lower the thermal conductivity, the slower the temperature change. Therefore, this quantity is generally evaluated in works involving PCM via non-stationary thermal conductivity tests. Obviously, with the incorporation of PCM, it is desired that the mechanical characteristics of asphalt mixtures are at least conserved at acceptable levels. In this sense, it is vital to carry out detailed studies, and the mechanical properties of pavements incorporating PCM must be deeply considered. For this purpose, wheel tracking and dynamic creep tests are most suitable for examining asphalt pavement’s permanent deformation [24,25,26]. 

Building upon the rich tapestry of research by Anupam and colleagues [27], who expertly discussed a variety of remarkable pavement technologies, encompassing reflective, evaporative, and innovative PCM-incorporated pavements, this work ventures further into the developmental landscape of PCM applications in asphalt pavements. However, it is pertinent to clarify the terminology and its context in these advancements.

“Cool pavement technologies” refer to a collective term for various engineering methodologies designed to reduce the urban heat island effect by manipulating the thermal characteristics of pavements. These methods can include using reflective surfaces, evaporative materials, or heat storage modification strategies, each with strengths and challenges.

Reflective pavements, for instance, often use lighter coloured or high albedo materials that can reflect a more sizeable portion of incident solar radiation, thereby absorbing less thermal energy and staying more relaxed. On the other hand, evaporative pavements utilise water retention and evaporative cooling, benefiting from heat absorption during the liquid–vapour phase transition of water.

The lesser explored and still evolving concept is the modification of heat storage properties. This involves manipulating the specific heat capacity of pavement materials, often through the incorporation of PCM. This exploration aims to detail a bibliometric review and systematic analysis of recent research outcomes incorporating PCM in asphalt pavements. It offers a comprehensive perspective on pavement performance, considering a myriad of aspects such as PCM type, particle size, melting point, enthalpy of fusion, leakage prevention strategies, PCM incorporation methods into asphalt mixtures, the percentage of PCM composite incorporation, and their thermal and mechanical performance.

The primary aim of this research lies in presenting a pioneering systematic and bibliometric evaluation in the realm of phase change materials (PCM). This analysis offers unparalleled depth and breadth, standing out as a unique endeavour in the field. The study intends to explicitly:Highlight the most prevalent PCM types used in the industry;Explore specific details, such as PCM particle sizes;Examine enthalpies of fusion and determine appropriate melting points;Discuss strategies to prevent PCM leakage;Detail the techniques for incorporating PCM in asphalt mixtures and shine a light on PCM composite incorporation;Thoroughly assess both the thermal and mechanical performance of PCM-infused pavements.

This effort not only strengthens the existing knowledge base in this area but also positions the insights and advancements from this study at the forefront of cool pavement technology research. By exploring various PCM types, their integration mechanisms, and the subsequent performance characteristics of PCM-enhanced pavements, this research opens many opportunities for addressing urban heat challenges. Moreover, this study sets the foundation for further refinement and deployment of this innovative technology by identifying future research paths and optimisation requisites.

## 2. Methods

### 2.1. Database

The organisation of this research utilised the Scopus database, chosen due to its multidisciplinary span, prominence, and recognition as one of the most extensive, international peer-reviewed databases [28]. The defined search strings included “Phase Change Material” OR “PCM” AND “asphalt” OR “bitumen,” applying these terms to probe article titles, abstracts, and keywords. Asphalt and bitumen were selected as synonyms to garner extensive literature on applying phase change materials in asphalt/bituminous binders and mixtures. The use of these terms acknowledged regional preference, with ‘asphalt’ typically employed by American researchers and ‘bitumen’ preferred by Europeans.

Furthermore, an integral aspect of this research was the execution of a systematic review. This methodical approach synthesises available evidence from relevant studies, enabling an in-depth understanding. This systematic review sought to explore the practical application of PCM in regulating pavement temperatures and mitigating the urban heat island effect. The review was guided by pertinent questions that underpinned the review process, elucidating the different aspects of PCM application in asphalt mixtures.

A standout feature of this study is the blend of a systematic review and bibliometric analysis, offering comprehensive insight into the literature on smart asphalt mixtures enriched with PCM. Only articles written in English and published from 2011 onwards were considered to ensure result accuracy. The bibliometric analysis conducted on 3 December 2022 identified 100 articles.

All 100 articles underwent a bibliometric review. For the systematic review, abstracts were read to identify articles addressing the urban heat island effect and pavement temperature reduction. This process refined the list to 45 relevant articles from the initial 100, ensuring alignment with the research objectives. Below, Figure 2 shows the flowchart to simplify the methodology used.

### 2.2. Bibliometric Approach

This study aimed to appraise the existing scientific landscape concerning asphalt mixtures integrated with phase change materials (PCM). Bibliometric analysis was employed to assess the available literature to achieve this objective. Our methodology was segmented into three stages: research preparation, data mining, and bibliometric analysis [29].

Research preparation: This was the initial phase in bibliometric analysis, which involved identifying and selecting appropriate databases and search terms relevant to the study topic. This stage aimed to collect as much relevant literature as possible to provide valuable insight into the subject matter.Data mining: After accumulating the research articles, the next phase, data mining, was commenced. This stage involved scrutinising the gathered data, including a manual review of the articles. The focus was on selecting articles that aligned with the research objectives, ensuring the consistency of the information used for analysis. Any potential duplicate documents were identified and removed at this stage. The data mining phase, therefore, refined the initial broad collection of articles into a more focused and relevant database for analysis.Bibliometric analysis: This involved using bibliometric tools to analyse and interpret the data obtained from the previous stages. In this analysis, we employed VOSviewer software (version 1.6.18), which enabled us to generate network maps that presented a visual representation of bibliographic coupling, co-authorship, and leading countries and institutions, among other things.

To proceed with the bibliometric analysis, the following research questions were defined:

How relevant is the theme of asphalt mixtures with PCM in the current literature? How have publications on the theme evolved?What are the most relevant terms identified in the co-occurrence network of this research scope?Which countries and institutions are most concerned about asphalt mixtures with PCM? What is the link between scientific research groups?Who are the most productive authors and co-authors in this research field?Which scientific journals are publishing the most on the theme?

### 2.3. Systematic Approach

A systematic review was carried out to explore the application of PCM in reducing pavement temperatures and mitigating the UHI effects. The methodology involved a thorough literature review employing a systematic approach to select review articles from the Scopus database, one of the most important databases in the world. In conclusion, the study provides insights into the potential of PCM in managing pavement temperatures and mitigating the UHI effect.

Which types of PCM are used? To prevent the UHI effect, what is the adequate melting point temperature for asphalt mixtures?What are the enthalpy of fusion values of PCM?What are the strategies used to prevent leakage of PCM in asphalt mixtures?Were there significant damages to the mechanical properties of asphalt mixtures incorporated with PCM?In comparative terms, what thermal improvement was achieved by incorporating PCM in asphalt mixtures?

## 3. Results

### 3.1. Bibliometric Analysis

Figure 3 depicts the temporal evolution of publications on asphalt mixtures with PCM. The Scopus database showed a significant increase in research on this topic since 2014, with between 2019 and 2022 witnessing the highest growth rate, accounting for over 80% of all publications. Notably, 2022 showed a peak in publications, with 22 articles. It is essential to highlight that this data was current as of 3 December 2022. These findings indicate that asphalt mixtures with PCM have emerged as a recent and innovative subject of scientific investigation in asphalt pavements.

#### 3.1.1. Main Terms

The index keywords that occurred at least five times are depicted in the network map shown in Figure 4. The size of each node is proportional to the term co-occurrence frequency. Meanwhile, the link line and distance between two distinct terms illustrate their relationships. Consequently, shorter distances and thicker link lines indicate higher co-occurrence. From these similarities and robust correlations, the terms were grouped into clusters. Interestingly, this network did not feature the term “UHI” (urban heat island). Despite its relevance to the subject matter, it was not a frequently co-occurring term in the selected studies, which adds an intriguing dimension to the analysis.

Central terms such as “phase change materials”, “asphalt mixtures”, “temperature”, and “heat storage” had the highest frequencies, and they were strongly connected with most of the other terms due to their high relevance for all clusters. There are no duplicated or similar terms on the map. The results were organised into four clusters by colour, as follows: (i) red—“differential scanning calorimetry”, “thermogravimetric analysis”, “pavement temperature”, “thermal performance”, and “atmospheric temperature”; (ii) green—“thermal conductivity”, “specific heat”, “cooling effects”, “solid–solid phase change material”, and “polyurethanes”; (iii) blue—“heat storage”, “latent heat”, “microencapsulation”, “scanning electron microscopy”, and “dynamic shear rheometer”; (iv) yellow—“enthalpy”, “binders”, and “rheological property.” The first cluster (i) was more related to laboratory tests, thermal analyses of pavement, and environmental effects of asphalt mixtures with PCM, while the second one (ii) had more terms linked to PCM type and their thermal properties. Keywords of the third cluster were mainly related to modified asphalt mixture performance related to latent heat thermal storage capability. Finally, the terms presented in the fourth cluster were related to asphalt mixture mechanical properties and the modification of binders.

Knowing the main keywords used in articles might be a helpful tool to make searching for articles easier according to the researcher’s interest. Moreover, it will help researchers select better keywords, making documents easier to find in databases such as Scopus.

#### 3.1.2. Co-Authorship

Figure 5 shows the results for the co-authorship network. The network map plotted articles with at least five co-authors, forming four clusters. The green cluster was composed of Chinese authors. Ma, B. can be considered the most productive researcher with 20 co-authored published essays and more than 330 citations, followed by Wei, K. with 14 papers and 160 citations. The primary common focus of these works by Ma, B. and Wei, K. was the application of PCM in asphalt mixtures, investigating their potential in temperature regulation and improving the overall performance and durability of asphalt pavements. The red cluster was composed mainly of Swiss authors, of which Partl, N.M. had 6 articles and was the most strongly connected.

It can be observed that the blue and yellow clusters present low numbers of co-authorship links. The possible cause is the high variability of co-authors in publications, i.e., a different research team wrote each article. Thus, they did not meet the threshold of 5 articles to appear on the network map.

In this database, 236 authors published about asphalt mixtures with PCM, among which the top five most productive single authors are detailed in Table 1. The authors were mainly from China and the same research organisation (Chang’an University), except for Partl, M.N. from Swiss Federal Laboratories for Material Science and Technology. Those institutions should be the current main centres of scientific knowledge on this theme.

Table 2 shows the five most relevant articles by their number of citations in the Scopus database. They had at least 50 citations each, and their focus was on assessing the feasibility and performance of different types of PCM incorporated into asphalt mixtures, such as graphite/polyethylene glycol composite or binary fatty acid with diatomite, for example. Also, they were mainly concerned with the thermal performance of binders or thermoregulation of asphalt mixtures. It is essential to highlight that the most cited articles were published by different research groups from China, Portugal, and the United States, among others.

The co-authorship database by organisation, with a minimum frequency of three citations, generated eight clusters (Figure 6). It is possible to observe that half of the clusters were isolated (working without collaboration among institutions), and the other half of the clusters had just one link. This indicated that cooperation between research groups still needs improvement regarding asphalt mixtures with PCM.

The Key Laboratory for Special Area Highway Engineering of the Ministry of Education (Chang’an University) and the School of Modern Post (Xi’an University of Posts & Telecommunications) were the Chinese research groups linked with the most significant number of partnership publications. Only one case connected groups from different nationalities: the College of Civil and Transportation Engineering—Hohai University (China) and the Ingram School of Engineering—Texas State University (USA). Therefore, there was a low geographic spread of investigations in addition to weak connections.

Figure 7 illustrates the co-authorship network map stratified by country, with each node signifying that researchers contributed to at least two articles. This underscored the global collaboration within the scientific community, as researchers, irrespective of nationality, continue contributing to the body of knowledge while working in various countries worldwide. Nine countries can be seen on the map, organised in five clusters, of which just two (green and red) were connected. The green cluster comprised China, Malaysia, and the United States; the red cluster included Canada, Sweden, and Switzerland. The size of each node confirmed that China was the most productive country, with 56 articles. It represented at least 74% of co-authorship publications about asphalt mixtures with PCM and had the strongest links to many other countries. Countries like the United States and Switzerland also had co-authorship articles representing relevant geographic locations in America and Europe, respectively, with 11 and 7 publications. The clusters represented by South Korea, Iran, and Poland had 2 articles, without connections.

This scene reinforced that incipient partnership research and published articles were concentrated in a few countries. China was the leader in asphalt mixtures with PCM research.

Figure 8 shows the bibliographic coupling network by source (journals) with at least two articles published. Indeed, 9 journals appear on the network map, and the range of publications was between 2 and 37. The most important journal was *Construction and Building Materials*, with 37 published articles representing almost 50% of all articles about asphalt mixtures with PCM on the Scopus database and about 57.9% of the total citations registered (736). While the other journals still had an incipient number of published articles, *Materials* and *Journal of Materials in Civil Engineering* had 6 and 4 articles, respectively. The main knowledge areas of the sources were civil engineering, materials science, and energy.

### 3.2. Systematic Review

The following sections encapsulate data collated from 45 documents that expounded upon the application of PCM in binders or asphalt mixtures. The analysis scrutinised nine performance aspects: PCM type, particle size distribution, melting point, enthalpy of fusion, strategy to prevent leakage, method of PCM incorporation in asphalt mixtures, percentage of PCM composite incorporation, as well as the thermal and mechanical properties of asphalt mixtures. Comprehensive summarisations of the systematic review’s findings are presented in Table A1, Table A2 and Table A3 located in Appendix A.

#### 3.2.1. Phase Change Materials Properties

Phase change materials (PCM) encompass a broad group, with diverse types such as specific waxes, including ceresin [30], and substances like n-tetradecane [31]. The diversity extends to compounds, such as stearic and palmitic acids [32]. Despite being less commonly used, including stearic and palmitic acids stems from their potential effectiveness in specific applications [33]. With their high melting points and enthalpies of fusion, stearic and palmitic acids are particularly suitable for applications with high temperatures and significant thermal energy storage [34]. These properties are detailed in Table A1, demonstrating their necessary roles in various applications. Also, as shown in Table A1, the most frequently employed materials as PCM are polyethylene glycols, or PEGs, notably PEG2000 and PEG4000, due to their significant enthalpies during phase changes [12,13,25,35,36,37,38,39,40,41,42,43]. These materials exhibit desirable attributes such as thermal and chemical stability, non-toxicity, and an appropriate phase change temperature range [44] during the phase change from solid.

The particle sizes of PCM, an aspect key to leakage risk (pointed out in Table A2) and thermal performance, have been studied extensively. The PCM in Table A1 exhibit a wide range, from a minuscule 0.0002 mm [45] to a larger 9.5 mm [37,46], with an average particle size of 1.27 mm. In the case of common materials such as PEG2000 and PEG4000, the average particle sizes are 0.4 and 0.6 mm, respectively. Interestingly, the size of the PCM particle impacts the leakage risk and thermal performance. Due to their greater surface area-to-volume ratio, smaller particles can enhance the thermal performance of PCM [47]. However, they are more susceptible to leakage as they can more easily navigate the carrier matrix material. Larger particles, on the other hand, present a reduced risk of leakage but might compromise the PCM’s thermal efficiency [47]. Figure 9 elucidates the interplay between PCM particle size, distribution, and consequent impacts on thermal performance and leakage tendencies. In Figure 9A, the dynamics of smaller PCM particles are highlighted. Their inherent surface-to-volume ratio advantage allows for homogenous dispersion across the material matrix. This even spread amplifies the potential for thermal regulation, moderating the overall temperature dynamics of the medium. Nevertheless, a trade-off exists: the facile movement of these smaller particles through the matrix augments leakage risk. Conversely, as portrayed in Figure 9B, larger PCM particles follow a contrasting behaviour. Their distribution is less uniform, leading to pockets of material that might be devoid of the thermal modulation benefits offered by PCM. While the silver lining with these larger particles is their reduced inclination towards leakage, they pose a challenge in achieving the optimal thermal proficiency desired by incorporating PCM.

Following the thermal properties subject, two thermal properties are highlighted for PCM to work well: the melting point and enthalpy of fusion, shown in Table A1. The melting point determines the temperature range over which PCM can store or release heat [48]. Therefore, this indicates the temperature at which PCM will change their phase. The melting point of a PCM must be carefully chosen for the specific application. If applied with high temperatures, a PCM with a low melting point may become unstable and decompose, melting quickly, and may not be able to store enough heat [49]. However, for the same application, a PCM with a high melting point may not be compatible with some materials. For instance, for a plastic that can melt at a lower temperature, the PCM will only melt once the temperature is too high [50]. Therefore, the melting point of a PCM determines its thermal stability and potential applications, alongside other factors like its chemical composition, molecular structure, and particle size, in deciding how and where a PCM can be effectively utilised. The PCM in Table A1 exhibit a vast array of melting points, ranging from as low as 3 °C, as in the case of R3 rubitherm RT [51], to as high as 80 °C, as with stearic acid [11]. The average melting point across all materials stands at 46.67 °C.

In Table A1, certain PCM, such as small-molecule alkane [36], n-tetradecane [31,52], PEG 800 mixed with polyacrylamide [53,54], solid–solid PCM (provided by a commercial company) [55], silica with ethyl cellulose [56], ethyl cellulose [57], four different types of composite phase change materials (CPCM) [58], and tetradecane (n-alkane C14H30) [59], have melting points around 20 °C or lower, which could make them not particularly effective in asphalt temperature regulation to mitigate the UHI effect. A PCM with a melting point around 20 °C would not be suitable for applications in which the temperature can reach 60 °C. This is because the PCM would melt at 20 °C, and then it would not be able to store any more heat [60]. If the temperature continued to rise, the PCM would start to release heat, but it would be unable to hold any more heat. This could be a problem in applications in which keeping the temperature within a specific range is essential. If that PCM was used for asphalt, it would not be able to store any more heat once the temperature reached 20 °C. This could lead to overheating. In that situation, when a PCM melts, it undergoes a phase change. This phase change can be exothermic, releasing heat [61]. If the temperature of the PCM is already high, the exothermic phase change could cause the temperature to rise even further and increase the UHI effect [61].

Conversely, PCM with higher melting points around 50 °C to 60 °C, such as types PEG1000 to PEG8000 [12,13,25,26,35,36,37,40,42,62], OP55E [63], stearic acid (SA) [11,64], PCM-43 and PCM-48 [46], and ceresin wax [65], could be employed in high-temperature thermal energy storage applications. These are particularly suitable for tropical and temperate locales where summer temperatures can get excessively high, posing a risk to the mechanical integrity of asphalt material. These PCM can slowly absorb and store excess heat during the hottest hours of the day. This makes them well-suited for applications in which keeping a large amount of thermal energy is essential, helping to reduce asphalt temperatures and mitigate the formation of UHIs [66]. They are stable at temperatures up to 60 °C or higher. This means that they can be used in applications in which the temperature may occasionally exceed 60 °C without the risk of the PCM becoming unstable or decomposing. This makes them versatile materials for use in various applications [66].

For materials like PEG2000 and PEG4000, a higher melting point indicates superior thermal stability. This makes them useful in specific high-temperature applications, such as industrial processes operating at elevated temperatures to store excess heat. PEG4000, with an average melting point of 56.14 °C, provides enhanced thermal stability over PEG2000, with a slightly lower average melting point of 50.63 °C [12,13,25,35,36,37,38,39,40,41,42,43]. These insights into melting points and their application in cooling asphalt reflect the fundamental role of PCM in mitigating the UHI effect.

The enthalpy of fusion, measured in joules per gram (J/g), which is the energy required to change a material’s phase from solid to liquid without changing its temperature, is one of the most significant properties of PCM. It determines the thermal energy the material can store and release during phase transitions [67]. In this work, the average values were between 119.73 J/g for the specific application of mitigating the UHI effect using PCM incorporated into asphalt pavements, and the materials of interest had high fusion enthalpies [37]. Materials with high fusion enthalpies can store and release more thermal energy during phase transitions, providing more significant potential for thermal regulation [68]. Looking at Table A1, a viable choice requires a high fusion enthalpy to store more energy and a melting point near ambient temperature to ensure the material will change phase when needed. To meet these demands, the following materials are good possibilities for some applications:Stearic acid (SA) [11,32,64,69,70]—enthalpy of fusion = 159.6–221.6 J/g; melting point = 40–80 °C. It has a high fusion enthalpy, and the melting point is within a reasonable range for different climates;PCM-43 [46]—enthalpy of fusion = 210 J/g; melting point = 43 °C. This material has an even higher fusion enthalpy, but its melting point is high. This may make it more suitable for hotter climates with higher ambient temperatures;n-tetradecane [31]—enthalpy of fusion = 256.2 J/g; melting point = 17.1°C. This material has the highest fusion enthalpy of the provided examples, but its melting point is not so high. This means it might not be the best choice in hotter climates with ambient temperatures above or approximately 17.1 °C.

As established in Table A1, stearic acid (SA) [11,32,64,69,70] and PCM-43 [46] appear to be the best candidates, with high fusion enthalpies and melting points that could be suitable for many climates. Notably, the average enthalpy of fusion values for PEG2000 and PEG4000 are 180 and 135.2 J/g, which fall outside the range of temperatures experienced by asphalt pavements and would not be so effective. However, as was detailed above, the melting points for these two types of PEG are suitable for applications in asphalt to mitigate the UHI effect, highlighting their potential as PCM [23]. Coupled with other favourable properties, such as low toxicity, high thermal stability, biodegradability, and a suitable melting point, PEGs are ideal for various thermal energy storage applications [46,47,48].

Despite the diverse applications of PCM, questions remain about their real-world application at high temperatures and significant thermal energy storage. Even though data, such as in Table A1, shed light on their roles in various contexts, a more detailed examination is essential. While certain materials capture attention due to their remarkable enthalpies during phase changes, attributes like thermal and chemical stability and non-toxicity are noteworthy. However, the relationship between these properties and their practical application needs to be clarified and requires further investigation.

The particle size spectrum for PCM extends from an almost microscopic 0.0002 mm to a noticeable 9.5 mm. While smaller particles are believed to boost the thermal efficiency of PCM, they might also escalate leakage risk. Conversely, more considerably sized particles could reduce leakage concerns but adversely affect thermal efficiency. The balance between these potential advantages and disadvantages has yet to be fully understood and necessitates explicit research.

The thermal properties of PCM, particularly the melting point and enthalpy of fusion, are essential. The melting point range is vast, indicating the potential for diverse applications. Nevertheless, there is a noticeable gap in the need for precise guidelines on the appropriate selection and use of these materials for specific applications. Some PCM seem less suited for temperature regulation, while those applicable for high-temperature thermal energy storage still need to be thoroughly understood in real-life high-temperature conditions. The practical applications and resulting efficacy for many materials, even those with promising characteristics, have yet to be definitively established.

The optimal fusion enthalpy for specific applications, like mitigating the UHI effect in asphalt pavements, must be clarified. While PCM show promise in addressing the urban heat island effect, the intricacies of their application still need to be explored. As urbanisation and climate challenges continue to intensify, realising the full potential of PCM mandates a more profound and systematic research approach.

#### 3.2.2. PCM Incorporation into Asphalt Mixtures and Leakage Prevention

In addition to the thermal properties depicted in Table A1 and Table A2, this study presents a thorough review that highlights the diversity of strategies employed to prevent the leakage of phase change materials (PCM) in asphalt mixtures, the methods for their incorporation, and the percentage of PCM composite used.

The propensity for leakage is one concern addressed in Table A2. The practical application of PCM is intricate due to their susceptibility to leakage during phase transitions. Ensuring the integrity of these materials, hence averting potential losses, represents a significant challenge in PCM selection and use, demanding an array of strategies [71]. PCM, varying in their composition and application, necessitate different methods and coating materials. For example, a PCM used for heat storage in buildings may require a carrier matrix that excels in preventing leakage [72], whereas a PCM employed in asphalt mixtures should not only lack leakage but also withstand high temperatures during the mixing process [73]. For vehicular applications, a carrier matrix that is both lightweight and robust might be more suitable [61].

A commonly adopted solution to prevent leakage involves using carrier matrix materials, such as SiO_2_, carbon-based materials, and diatomite [25,26,38,41,42,43,44,62,64,69,74,75,76]. These substances serve as physical barriers, encapsulating the PCM within the system to minimise leakage. For instance, SiO_2_ encapsulation is frequently used with popular PCM like PEG2000 and PEG4000 [25,38,41,42]. Additionally, there are less commonly used strategies, such as adsorption on diatomite silica microporous structures [62,64], the use of PCM in composite form [38,58], polymerisation [31,77,78], emulsion polymerisation [52], and integration with multi-walled carbon nanotubes (MWCNT) [45], along with more advanced techniques like microencapsulation [79] and shape-stabilised expanded graphite [26,80]. Carrier matrix materials can also influence the thermal properties of PCM under various conditions (as shown in Table A1). They can enhance the PCM’s thermal conductivity, promoting efficient heat storage and release. Furthermore, the carrier matrix can protect the PCM from degradation by shielding it from environmental factors such as air and moisture, thereby reducing the maintenance costs of PCM [81]. The matrix can also regulate interactions between the PCM and its surroundings, serving as a physical barrier that may delay melting of the PCM by limiting its exposure to certain temperatures [82].

Certain materials are inherently solid and do not necessitate a leakage prevention strategy. Solid-to-solid PCM (SS-PCM) are a unique class characterised by their ability to transition from one solid phase to another [40,55,63,75]. Unlike other PCM, they remain solid throughout their phase transitions, eliminating the need for leakage prevention strategies. This property makes SS-PCM exceptionally suitable for various thermal energy storage applications due to their reversibility [83]. There are two principal types of SS-PCM, differentiated by their molecular interactions. The first type involves the rearrangement of molecules from one crystalline phase to another [84,85,86]. This structural transformation changes the thermal characteristics of the material without changing its physical state. The second type involves incorporating crystallisable sections through chemical bonding into a secondary structure. This change can influence the material’s thermal properties, including enthalpy, transition temperature, and thermal conductivity [77,83,85,87]. Figure 10 provides a visual schematic of the phase change processes in SS-PCM systems. In part (A), the transition from an ordered crystalline phase to a disordered non-crystalline phase is vividly depicted for the system. This transformation underlines the inherent ability of SS-PCM to undergo structural rearrangements. Part (B) shows the SS-PCM transition from a neatly arranged, regular crystalline structure to a more chaotic and random non-crystalline phase. This change is pivotal in understanding the thermodynamic properties of SL-PCM systems and their potential applications in energy storage and release.

Regarding their workability, SS-PCM are considered easy to handle due to their solid state. However, the specific characteristics of these materials can vary widely, influencing their ease of use [88]. Some SS-PCM may require special handling or incorporation methods to achieve the best performance. For instance, certain SS-PCM may need to be mixed with other materials to improve their thermal conductivity or stability [89]. Nonetheless, this is due to their solid state and lack of leakage [90].

PCM with specific encapsulation strategies and careful selection based on their thermal properties can be beneficial in storing and releasing heat during peak temperatures, thereby reducing surface temperatures and the overall urban heat island effect. Solid-to-solid PCM (SS-PCM), due to their inherent solid state and absence of leakage, present a particularly advantageous option for urban heat mitigation. Due to molecular rearrangements or secondary structure modifications, their tuneable thermal properties make SS-PCM a versatile choice for diverse urban applications, offering a sustainable solution to urban heat islands.

Another point addressed in Table A2 is the most used method to incorporate PCM into asphalt mixtures. The addition of PCM is typically achieved via two methods: wet and dry mixing. The wet mixing method combines the PCM and the binder, providing a simple and straightforward approach to integrating PCM into mixtures [91]. This method’s simplicity has made it the most prevalent technique in the field [12]. On the other hand, the dry mixing method involves the addition of the PCM during preparation of the binder [92]. This method, although slightly less common, is gaining traction due to its enhanced control over the distribution of incorporated materials, like PCM, within asphalt mixtures [93]. Both methods have their unique advantages and potential drawbacks. Wet mixing is often seen as more efficient since it eliminates the need for additional steps [94]. However, this method might need to be revised to control PCM distribution throughout the asphalt mixture, potentially affecting its performance. Conversely, while the dry mixing method allows for improved control over PCM distribution, it can be more laborious and might present challenges in ensuring a homogeneous mix [95]. These two techniques can be refined with a focus on efficiency and effectiveness for better PCM incorporation.

Once the best method to incorporate the PCM in the asphalt mix has been defined, the quantity of PCM to be added must be known. This can be achieved through volume or mass substitution. However, analysis of Table A2 shows that there is no one-size-fits-all solution, as the choice of method often depends on the type of PCM used and the desired thermal properties of the final product. Substitution can replace either the volume or mass of the aggregate or binder utilised [35,37,39,46,53]. 

Specific studies have implemented PCM contents ranging from 3% to 15% in terms of either volume [38] or mass [32,63,70,79] of the binder in the asphalt mixture. Some unique combinations have been made by substituting 50% of the fine aggregate with sizes of 0.3–0.6 and 0.15–0.3 mm [69]. Across the board, the percentages of substitution most often seen in the literature range from 2% to 20% for both mass and volume substitutions of the binder. However, the frequency of these percentages varies widely according to the specifics of each experiment or study.

Further considerations reveal that the type of PCM used can affect the amount integrated into an asphalt mixture. Certain PCM possess higher latent heat than others, which can store more heat per unit volume. Consequently, less PCM would be needed to achieve a given thermal performance. The target thermal performance also plays a critical role in determining the amount of PCM required [11,32,64,69,70]. If the objective is to lower the temperature of the asphalt mixture by a few degrees, less PCM is needed as compared to a goal of significantly reducing the temperature.

Moreover, the equipment can also influence the amount of PCM incorporated into an asphalt mixture. Some devices might not be designed to accommodate large quantities of PCM, limiting the maximum amount of PCM that can be integrated into the asphalt mixture. This necessitates carefully evaluating the available resources when planning the incorporation of PCM.

The method of PCM incorporation into the asphalt, whether wet or dry, can be optimised based on the desired thermal performance and application specifics. The quantity of PCM used can be tuned to achieve targeted thermal properties, thus tailoring the urban surface to withstand intense heat conditions.

While methods to prevent leakage from PCM have evolved, they often demand specific solutions tailored to PCM composition and intended application. A universal strategy still needs to be discovered, offering a rich avenue for future exploration and innovation. Furthermore, even though carrier matrix materials effectively encapsulate PCM, their nuanced effects on PCM thermal properties still merit deeper investigation. Notably, solid-to-solid PCM (SS-PCM) provide a promising leakage-free solution, yet their varied characteristics demand a better grasp to leverage their full potential. The debate between wet and dry mixing methods for PCM incorporation still needs to be solved, with neither proving definitively superior. The optimum PCM quantities in asphalt mixtures must also be revised, with decisions often hinging on the PCM type and desired thermal attributes. Intriguingly, the equipment used in the process can sometimes cap the amount of PCM integrated, suggesting potential technological advancements. Lastly, the method chosen for PCM incorporation demands careful optimisation to maximise benefits aligned closely with project objectives. All of these research gaps underscore the vibrant, evolving landscape of PCM integration into asphalt, offering many opportunities for future studies.

#### 3.2.3. Analysis of Thermal and Mechanical Characteristics in Asphalt Mixtures with Diverse PCM Varieties

Integrating PCM into asphalt mixtures, a significant development in materials science, must consider various aspects, including potential impacts on mechanical properties and the degree of thermal improvement realised [25,36,37,38,77]. The data in Table A3 reveal that these materials effectively manage thermal energy absorption and release, thus regulating temperature fluctuations in asphalt pavements [52,78,96,97].

The extent of this thermal regulation largely depends on the PCM content, with a higher dosage often resulting in more substantial temperature decline. Many factors, including ambient temperature [13,76], PCM type and dosage [26], and pavement structure [78], also influence the effectiveness of PCM usage in asphalt mixtures. These substances can also elevate the asphalt mixture’s albedo and specific heat capacity due to their augmented thermal conductivity and latent heat storage capacity [64,75]. Lab and field trials have indicated that PCM can lower pavement temperatures by as much as 9 °C [31,40,62,98].

However, from a mechanical perspective, incorporating PCM into asphalt mixtures may occasionally induce adverse effects. There is a tendency for dynamic stability to decline in modified mixtures with increasing PCM content [25], while numerous studies have noted a decrease in mechanical strength following PCM integration [37]. The lesser mechanical strength of PCM, compared to typical aggregates like limestone or basalt, can be an attributing factor for the reduced strengths of mixtures [37].

Moreover, strength evaluations from uniaxial compressive and splitting strength tests have revealed that strength parameters tend to diminish with increasing PCM content, especially when PCM have larger particle sizes. The overall strength of the asphalt mixture is less impacted when the PCM has a smaller particle size. However, even with these reductions, modified asphalt mixtures can enhance the anti-slip properties and durability of coated pavements [25,37]. Notably, several studies have highlighted improvements in specific areas of mechanical performance, such as an increase in the asphalt mixture modulus, contributing to better anti-rutting capability [36,38,76].

Conversely, the impacts on mechanical properties are inconsistent and vary based on the specific PCM used. For example, polyurethanes with a higher isocyanate content are associated with small phase change enthalpies and can withstand elevated temperatures without liquid leakage [77]. This suggests that the choice of PCM and its properties can significantly affect the mechanical performance of modified asphalt mixtures.

On the thermal optimisation front, PCM have proven to be effective in enhancing the thermal performance of asphalt mixtures. Pavements treated with PCM have demonstrated a significant cooling effect. During a summer day, the peak temperatures of both top and bottom surfaces can decrease by approximately 6°C to 8 °C [64] in samples. Furthermore, an optimal 14% (by mixture weight) PCM composite content can result in temperature modulation, achieving a temperature difference of around 9.0 °C (Figure 11) in asphalt specimens, as opposed to a reduction in global temperature [62].

The type of PCM also plays a substantial role in thermal improvement. For instance, the phase change temperatures of PEG2000 and PEG 4000 align more closely with peak pavement temperature (around 60 °C on summer days). As a result, these PCM can more efficiently regulate pavement peak temperature compared to PEG1000 [35]. Simulation results indicated that a PCM synthesised from PEG2000 (alkane) was suitable for controlling the temperature of asphalt mixtures, in both summer and winter conditions [36].

The type and quantity of PCM used are fundamental in determining the mechanical and thermal performance of modified asphalt mixtures. Despite challenges associated with mechanical performance upon PCM incorporation, the significant thermal improvements suggest that PCM hold immense potential for mitigating high-temperature effects on pavements [65]. However, further research and optimisation are necessary to devise the best strategies for PCM incorporation that yield the most effective mechanical and thermal performance.

Phase change materials (PCM) uniquely manage the thermal energy dynamics within asphalt pavements. They regulate temperature swings and provide crucial benefits in addressing urban heat islands. Their effectiveness, however, hinges on various factors, with the PCM concentration playing a pivotal role. Notably, a higher dosage of PCM may result in a more significant decrease in pavement temperatures, helping to reduce the urban heat island effect. Field and lab-based experimental trials have established the capability of PCM to decrease pavement temperatures, offering new prospects in urban heat island mitigation. However, the implications on the physical properties when incorporating PCM into asphalt mixtures must be considered. This introduction could undermine the mechanical resilience of asphalt mixtures, negatively affecting pavement durability. Balancing thermal benefits against potential mechanical compromises is critical. Optimising thermal improvements without jeopardising pavement integrity demands thorough planning and ongoing research. While the challenges are not insignificant, the substantial thermal benefits linked with PCM usage underline their significant potential in mitigating the urban heat island effect. The focus remains on the strategic deployment of PCM in asphalt mixtures, ensuring thermal control for pavement surfaces that contribute to urban heat island effect mitigation.

Considering the current understanding, several research avenues beckon for exploration to unravel the intricacies of PCM integration into asphalt mixtures. The variability in mechanical properties based on the specific PCM type needs elucidation to provide clarity on their impacts. A comprehensive investigation into how particle size directly influences asphalt mixture strength is essential to effectively tailor applications.

Though promising, the asserted optimal 14% PCM content beckons for a more comprehensive evaluation across varied climatic conditions and pavement types. Delving deeper into a comparative study between different PCM types will foster a nuanced understanding of their relative efficiencies, setting the stage for more refined applications. Strategising PCM incorporation, with a keen eye on maximising mechanical and thermal performance, remains a pivotal research frontier.

As the urban heat island effect becomes increasingly pronounced, leveraging PCM for optimal mitigation is significant. Insight into striking the right balance between thermal benefit and mechanical integrity will determine the success of PCM-integrated asphalt mixtures. While the current trajectory shows promise, meticulously exploring these research gaps will be instrumental in harnessing the full potential of PCM in asphalt pavements.

## 4. Conclusions

This comprehensive bibliometric and systematic review synthesises recent research on implementing PCM in asphalt mixtures, highlighting the significant contribution of this technology toward mitigating the UHI effect.

The evolution of the theme demonstrates that asphalt mixtures with PCM have emerged as a significant and burgeoning subject within the current scientific literature. A marked uptick in research on this topic, particularly between 2019 and 2022, accounting for over 80% of all publications, implies that the application of PCM in asphalt mixtures is a promising and innovative field of study within asphalt pavement engineering.

The most important terms identified within the co-occurrence network included “phase change materials”, “asphalt mixtures”, “temperature”, and “heat storage.” These terms were heavily interconnected and highly relevant across all clusters, indicating the primary focus areas within this research scope. However, the absence of the term ‘Urban Heat Island (UHI)’, despite its relevance, presents a potential research gap for future exploration.

China, notably Chang’an University, has emerged as the primary research hub for asphalt mixtures with PCM, contributing a substantial portion of the literature. The most productive authors in this field predominantly hailed from China, with their primary research focus on PCM applications and performance in asphalt mixtures. Their contributions to the body of knowledge have provided invaluable insight into this emerging research area.

The concentration of publications in a limited number of journals, primarily *Construction and Building Materials*, which published almost 50% of all articles about asphalt mixtures with PCM in the Scopus database, illustrates the need for broader dissemination of research findings across various platforms to ensure comprehensive coverage and accessibility.

Various PCM, such as paraffin waxes, fatty acids, and salt hydrates, have been utilised in asphalt mixtures. However, polyethylene glycols (PEGs), specifically PEG2000 and PEG4000, emerged as the most frequently employed PCM. This is likely attributed to their suitable phase change temperature range, chemical stability, and non-toxicity.

The appropriate melting point temperature for PCM utilisation in asphalt mixtures depends on the specific application and environmental conditions. The melting points can range from 3 °C to as high as 80 °C. However, PEG2000 and PEG4000, with average melting points of 50.63 °C and 56.14 °C, respectively, demonstrate a wide range of suitability.

Values of the enthalpy of fusion, which indicates the PCM’s thermal energy storage capacity, vary considerably, ranging from 2.11 to 256.2 J/g. PEG2000 and PEG4000 exhibit average enthalpy of fusion values of 180 and 135.2 J/g, respectively, underpinning their substantial thermal storage capacities.

Various strategies have been employed to prevent PCM leakage in asphalt mixtures, which can compromise the material’s integrity. Encapsulation with SiO_2_ was the most common method. Other innovative methods include PCM adsorption on diatomite silica microporous structures, composite form PCM, and polymerisation.

Integrating PCM into asphalt mixtures has potential repercussions on their mechanical properties. Some studies reported a reduction in dynamic stability and mechanical strength with increasing PCM content. However, despite these reductions, the modified mixtures typically met the required performance standards, suggesting that the overall integrity of the pavements was maintained.

The thermal performance of asphalt mixtures, a primary objective of PCM integration, has shown substantial improvement. However, the degree of improvement is contingent on various factors, such as PCM type, dosage, and pavement structure. Significant reductions in pavement temperature by up to 9 °C have been reported.

In conclusion, the systematic incorporation of PCM into asphalt mixtures demonstrates a promising avenue for enhancing thermal performance and combatting the UHI effect. However, further research is necessary to optimise the balance between thermal improvement and mechanical performance, ensuring this innovative technology’s functional and environmental benefits.

However, this review has identified several areas for improvement in the current literature that deserve future attention. The mechanical resistance of PCM has yet to be thoroughly explored, and the use of PCM derived from recycled material remains uninvestigated. Moreover, the protection of PCM utilising recycled materials is another under-researched area. Notably, the absence of standardised tests assessing PCM leakage was an explicit limitation within the existing literature.

Notably, the review underscores conspicuous gaps in the extant literature. There is a lacuna regarding the lifecycle analysis of these materials. The prevalent application of PCM—whether encapsulated in spherical form or directly infused into pavements—suggests a particular orthodoxy in the methodologies. The impact of the encapsulation’s geometry on the asphalt’s mechanical durability remains relatively uncharted, hinting at a nuanced research dimension. How might the material’s shape influence its resistance, longevity, or heat retention? Such considerations, if delved into, could revolutionise the practical applications and efficacies of PCM in pavements.

This study may serve as a valuable guide for researchers navigating this dynamic field, providing a thorough understanding of current trends, key contributors, and potential research directions. Future studies should continue to monitor the progression of this research field, updating the bibliometric analysis as the literature expands and facilitating the identification of and addressing research gaps to further promote the application of PCM in asphalt mixtures.

## Figures and Tables

**Figure 2 sensors-23-07741-f002:**
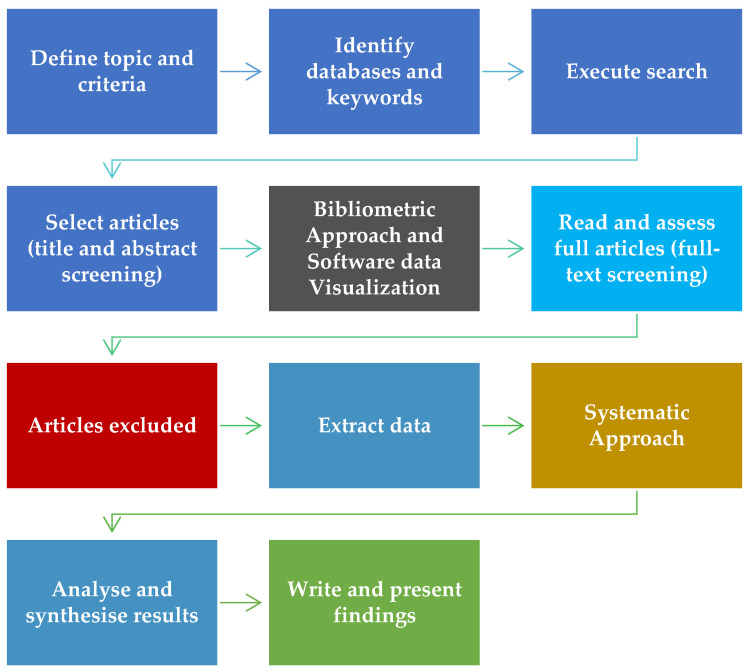
Methodology schematic chart.

**Figure 3 sensors-23-07741-f003:**
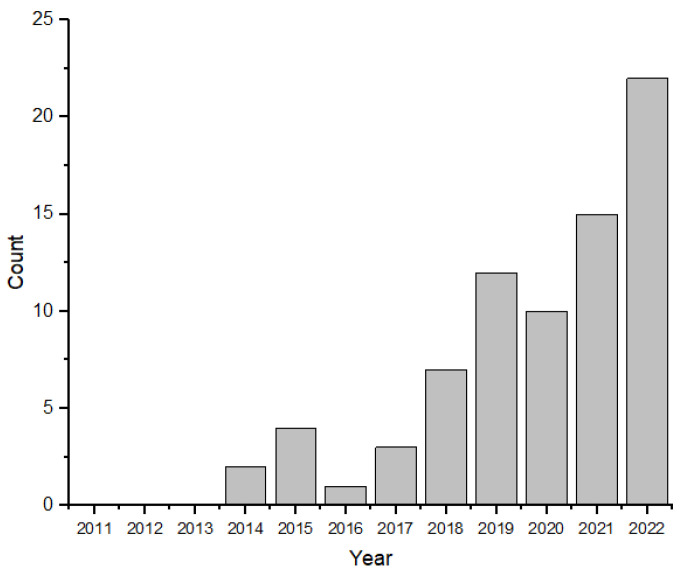
Number of scientific publications per year containing keywords (Scopus): “asphalt” and “phase change material” in the title and/or keywords of the article.

**Figure 4 sensors-23-07741-f004:**
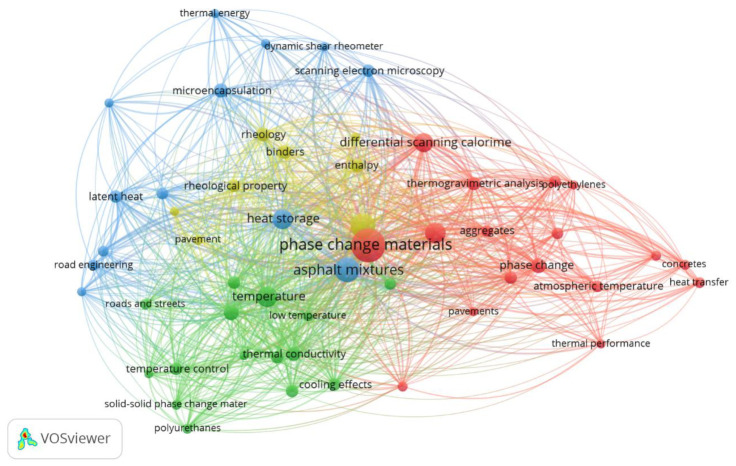
Co-occurrence network of the most frequent terms in titles, abstracts, and keywords for scientific articles published about asphalt mixtures with PCMs.

**Figure 5 sensors-23-07741-f005:**
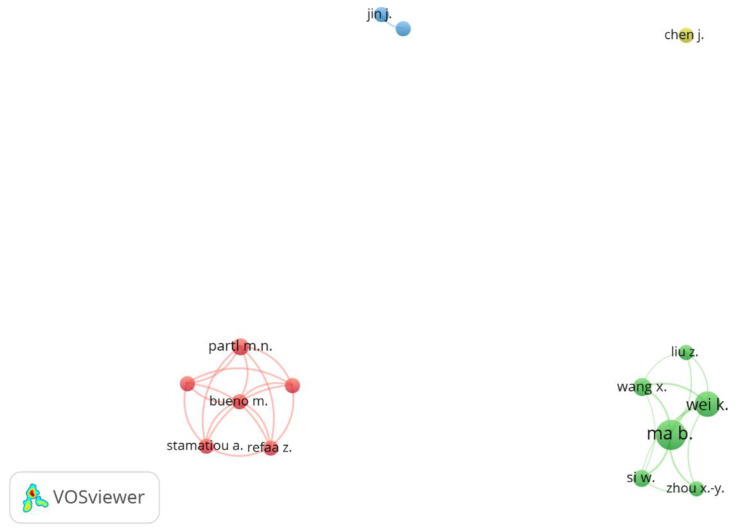
Network map of co-authorship by author.

**Figure 6 sensors-23-07741-f006:**
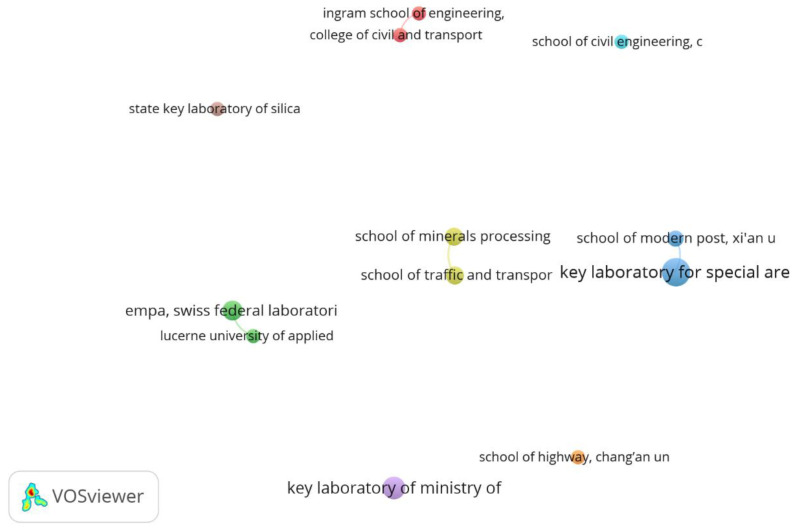
Network map of co-authorship by institutions.

**Figure 7 sensors-23-07741-f007:**
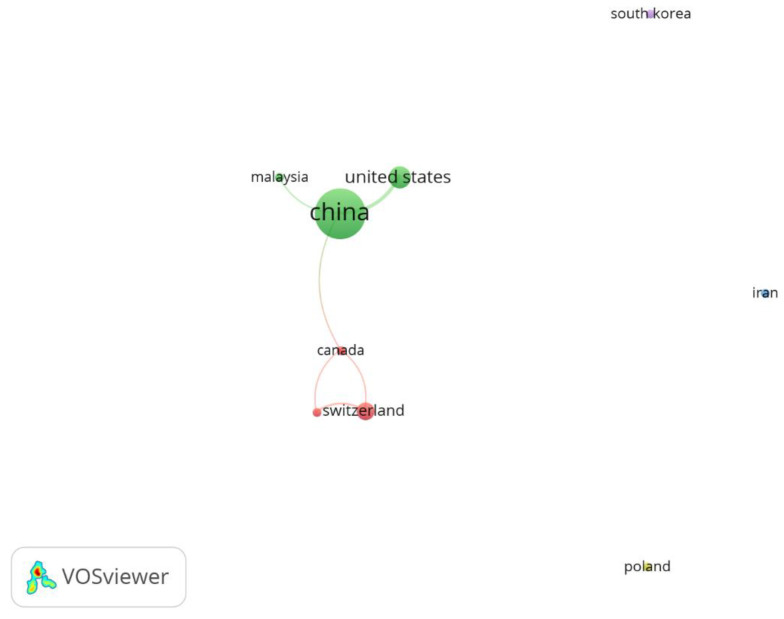
Network map of co-authorship by country.

**Figure 8 sensors-23-07741-f008:**
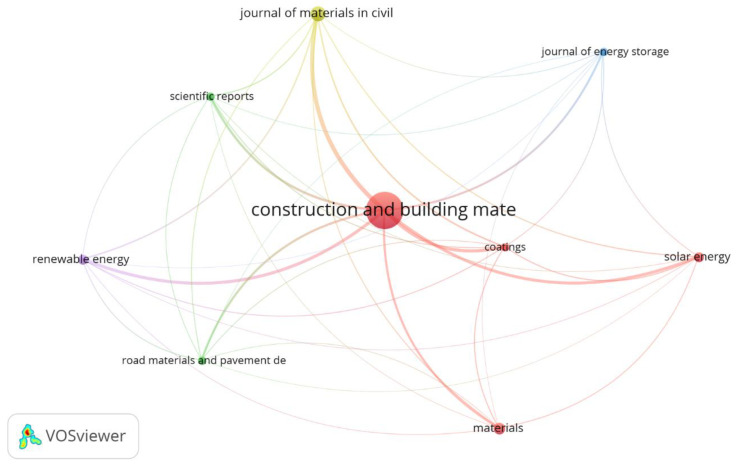
Bibliographic coupling by source.

**Figure 9 sensors-23-07741-f009:**
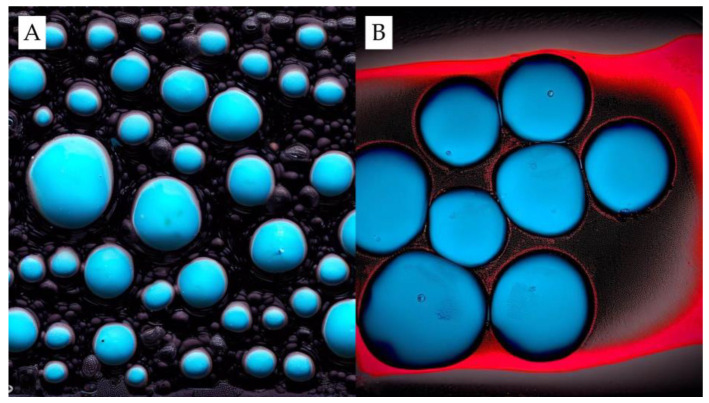
Distribution dynamics and performance impacts of PCM particle size in materials. (**A**) Majority of small particles and (**B**) without small particles.

**Figure 10 sensors-23-07741-f010:**
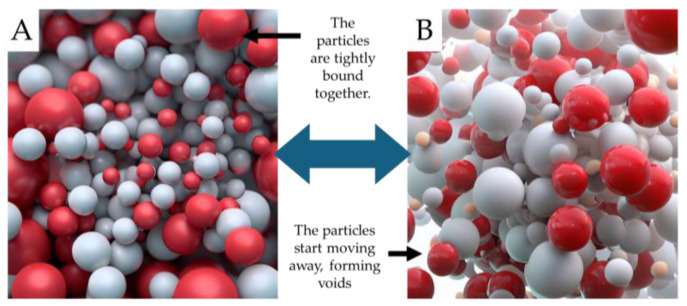
Phase transition dynamics in SS-PCMs. From (**A**) ordered crystalline to (**B**) disordered non-crystalline structures.

**Figure 11 sensors-23-07741-f011:**
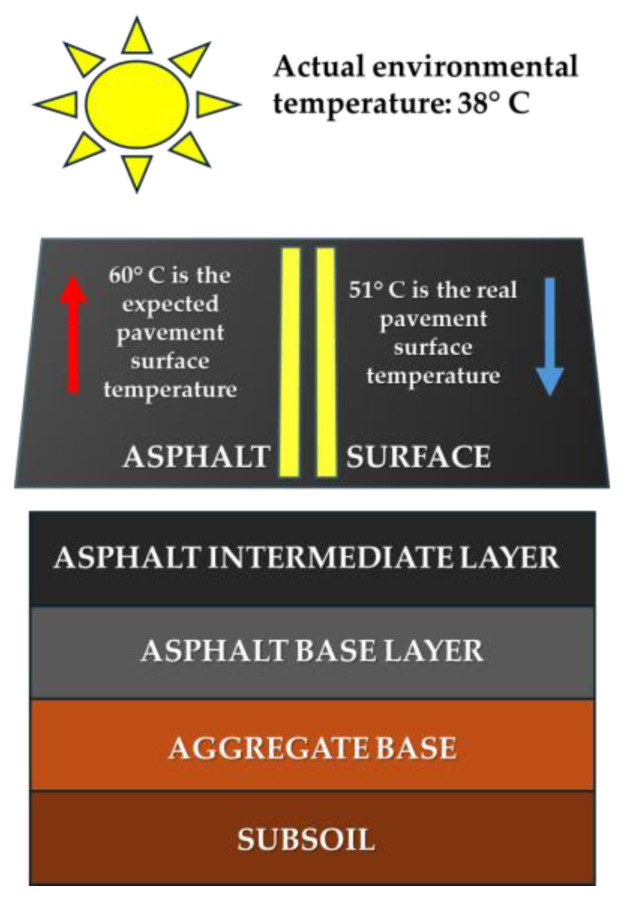
Layers of flexible pavement and the temperature difference in the pavement surface (9 °C).

**Table 1 sensors-23-07741-t001:** Most productive authors.

Author	Documents	Citations	Country	Affiliation
MA, B.	20	331	China	Chang’an University
WEI, K.	14	160		
WANG, X.	7	28		
PARTL, M.N.	6	146	Switzerland	Empa, Swiss Federal Laboratories for Material Science and Technology
SI, W.	6	155	China	Chang’an University

**Table 2 sensors-23-07741-t002:** Most cited documents in the Scopus database.

Title	Authors	Citations	Year
Assessing the feasibility of impregnating phase change materials in lightweight aggregate for development of thermal energy storage systems	Kheradmand, M., Castro-Gomes, J., Azenha, M., (…), De Aguiar, J.L.B., Zoorob, S.E.	73	2015
Performance analysis of incorporating phase change materials in asphalt concrete pavements	Athukorallage, B., Dissanayaka, T., Senadheera, S., James, D.	55	2018
Preparation of expanded graphite/polyethylene glycol composite phase change material for thermoregulation of asphalt binder	Zhang, D., Chen, M., Wu, S., Liu, Q., Wan, J.	52	2018
Preparation and thermal performance of binary fatty acid with diatomite as form-stable composite phase change material for cooling asphalt pavements	Jin, J., Liu, L., Liu, R., (…), Lin, F., Xie, J.	50	2019
Assessing the feasibility of incorporating phase change material in hot mix asphalt	Manning, B.J., Bender, P.R., Cote, S.A., (…), Sakulich, A.R., Mallick, R.B.	50	2015

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
