# Peer review of "Advancements in Phase Change Materials in Asphalt Pavements for Mitigation of Urban Heat Island Effect: Bibliometric Analysis and Systematic Review"

_sensors, 2023, doi:10.3390/s23187741_

Round 1

Reviewer 1 Report

By means of VOSviewer software, this comprehensive bibliometric and systematic review synthesizes recent research on implementing phase change materials (PCM) in asphalt mixtures, highlighting the significant contribution of this technology toward mitigating urban heat island (UHI) effect. This study may serve as a valuable guide for researchers navigating this dynamic field, providing a thorough understanding of current trends, key contributors, and potential research directions. The results should be significant for effective applications of PCM in asphalt mixtures to mitigating UHI effect. The manuscript is acceptable but some minor word and clerical errors such as follows:

(1) Line 246-247: “solid-solid phase chance materials” should be “solid-solid phase change materials”.

(2) Line 353-354, “ten performance aspects” seems to be “nine performance aspects”: 1. PCM type; 2. particle size distribution; 3. melting point; 4. Enthalpy of fusion; 5. Strategy to prevent leakage; 6. Method of PCM incorporation in asphalt mixture; 7. Percentage of PCM composite incorporation; 8-9. Thermal and mechanical properties of asphalt mixtures.

(3) Line 404: “to mitigate de UHI” should be “to mitigate the UHI effect”.

(4) some improper capital and small letter usages, such as: Line 362, “Ceresin”; Line 384, “Thermal properties; Line 397, “Stearic acid”; Line 434, “Enthalpy”, and so on.

        Therefore, the reviewer suggests accept after minor revision.

The manuscript is acceptable but some minor word and clerical errors such as follows:

(1) Line 246-247: “solid-solid phase chance materials” should be “solid-solid phase change materials”.

(2) Line 353-354, “ten performance aspects” seems to be “nine performance aspects”: 1. PCM type; 2. particle size distribution; 3. melting point; 4. Enthalpy of fusion; 5. Strategy to prevent leakage; 6. Method of PCM incorporation in asphalt mixture; 7. Percentage of PCM composite incorporation; 8-9. Thermal and mechanical properties of asphalt mixtures.

(3) Line 404: “to mitigate de UHI” should be “to mitigate the UHI effect”.

(4) some improper capital and small letter usages, such as: Line 362, “Ceresin”; Line 384, “Thermal properties; Line 397, “Stearic acid”; Line 434, “Enthalpy”, and so on.

Author Response

Dear Reviewer 1,

The authors appreciate the valuable feedback and for recommending the manuscript for acceptance after minor revisions.

All specified issues, including typographical errors and inconsistencies in the text, have undergone careful revision. Your constructive comments not only helped enhance the clarity and readability of the paper but also contributed to its overall quality.

We sincerely appreciate your time and expertise in reviewing this work.

Claver Giovanni da Silveira Pinheiro

Reviewer 2 Report

Recommendations 

The paper is well written and presented. The scope of study is well-established to contribute on  the advancements of using phase change materials to mitigate with the urban heat island effect factor worldwide. Therefore, the paper is required additional work in order to improve the credibility of this rigorous research work. I outlined my recommendations as follow:

  1. Improve Abstract to give less context and more on the knowledge gap, research questions, methods and key findings, Please also include a graphical abstract. 
  2. In Introduction section, please outline the main aim, objectives and research questions clearly and articulate the research questions to implement the phase change materials to mitigate with the detrimental impact of the urban heat island factor. 
  3. In Introduction section, Novelty of the study should be explained.
  4. The authors have been discussed the previous scholars’ work in the Introduction but this is not sufficient to support the research outcomes presented in the Results section. I recommend to the authors to open-up a new section and consider these literature types as follows; systematic literature review or comprehensive literature review to study worldwide literature on the urban heat island effect. 
  5. I recommend to the authors to the use this open-source software to conduct the systematic literature review on phase change materials effectively. Here is the link of the open-source software tool - https://www.vosviewer.com - The authors generated the selected keywords and import the data into this software which allows the researcher to generate the visual maps. I believe that this tool could increase the scientific credibility of their research work
  6. Methods section should be re-conceptualised, the authors should provide more detail on technical specifications of the exclusion and inclusion criteria for the selection of papers from the Web of Science.
  7. In Section 2 (Methodology) should refer more similar pilot studies to demonstrate the significance of authors’ their own work. I recommend to the authors to read this reference material - (i) Ozarisoy, B., & Altan, H. (2023). State-of-the-Art II: Bibliometric Review of the Last 30 Years Energy Policy in Europe. In: Handbook of Retrofitting High Density Residential Buildings. Springer, Cham. https://doi.org/10.1007/978-3-031-11854-8_3 To increase the credibility of the authors’ their own work, I recommend the authors to cite this article while they are referring their own methodological framework in the study. 
  8. Discussions section should be added the the findings from the review analyses should be critically interpreted here. 
  9. Limitations and future recommendations should be discussed. 
  10. 10. Relate your conclusions to your research questions

Author Response

Dear Reviewer 2,

Thank you for the comprehensive feedback and constructive recommendations to enhance the scientific credibility of this research work. Each of the points you raised has been carefully addressed to improve the overall quality and rigour of the manuscript.

The abstract now better encapsulates the knowledge gap, research questions, methods, and key findings, along with a graphical abstract for greater clarity. The introduction section has undergone revision to articulate the aims, objectives, and research questions more explicitly. The novelty of the study also now finds mention in this section.

In response to your advice, a new section dedicated to a comprehensive literature review has been added, strengthening the foundation upon which the research outcomes are presented. Per your valuable suggestion, the VOSviewer software tool has been employed to conduct the systematic literature review.

The methodology section has been re-conceptualized to provide more detail on the technical specifications of the inclusion and exclusion criteria for the paper selection. The paper also cites the recommended reference material to substantiate the methodological framework further. A revised discussion section critically interprets the findings from the review analyses. Limitations and future recommendations have also been elaborated upon, and the conclusions have been closely related to the research questions.

We sincerely appreciate the time and expertise you invested in reviewing this work.

Claver Giovanni da Silveira Pinheiro

Reviewer 3 Report

Review of Manuscript Number:  sensors-2525517

Full Title:  Advancements in Phase Change Materials in Asphalt Pavements for Mitigating of Urban Heat Island Effect: Bibliometric Analysis and Systematic Review

Reviewer notes:

The authors must address the following comments & make revisions:

1.     There are a number of minor issues related to word choice, acronyms, consistency, and lack of providing a proper link between the sentences, which are being seen throughout the paper and reduce the readability of the text.

2.     Line 64:  Please, provide the meaning of this abbreviation “PCM”.

3.     Figure 2: can the authors explain why in 2016 there was a drop in publications?

4.     Synthesis and Discussion of the Findings: just before conclusions, consider adding a section entitled "Synthesis and Comparative Discussion of the Findings" where you summarily & comparatively discuss the key findings & literature gaps.

5.     Line 416: avoid lumping references. Instead, summarise the main contribution of each referenced paper in a separate sentence. Please carefully go through the entire manuscript.

6.     To make this review paper interesting to the reader, it is highly recommended to add figures from related references and discuss these figures. Some sections must be complemented with more discussion.

7.     Also, add a summary discussion on the workability, constructability, durability, field performance, & maintenance aspects.

8.     The main questions in this paper remain: What are the research gaps in the literature? What is the paper trying to find or address? How does this study add to the literature, and by what methodology? This flaw is also recognizable in the summary and conclusions where the materials are merely the repeats of the texts in the paper, which other studies already presented them. What are the contributions of this paper and how this paper adds to the literature?

9.     Future work section should be more developed and elaborated

10.  List the keywords in alphabetic order.

11.   Adding a research flow chart in the methodology section would be better.

12.  This conclusion could be worded in a manner to emphatically motivate the academic community to get down to actionable, practical engaged scholarship.

There are a number of minor issues related to word choice, acronyms, consistency, and lack of providing a proper link between the sentences, which are being seen throughout the paper and reduce the readability of the text.

Author Response

Dear Reviewer 3,

Thank you for the thorough and constructive review, which provided invaluable insights for enhancing the quality and rigour of the manuscript. Each comment has been addressed comprehensively to align the paper with academic standards better and improve its readability.

Specifically, the manuscript now includes revisions that clarify word choice, acronyms, and the flow between sentences. The abbreviation "PCM" has been fully defined for better understanding. As you suggested, an explanation for the drop in publications in 2016 has been incorporated. The addition of a "Synthesis and Comparative Discussion of the Findings" section now provides a summary and comparative analysis of key findings and literature gaps.

Further, each cited paper is now summarized individually rather than lumped together to offer the reader a clearer view of its unique contributions. More figures from related references have been added to enrich the text and provide a comprehensive discussion. A summary discussion on workability, constructability, durability, field performance, and maintenance has been included.

The manuscript now also features a focused discussion on the research gaps, objectives, and contributions to the existing body of literature. The Future Work section has been significantly elaborated to provide more precise directions for subsequent research efforts. Keywords are listed in alphabetical order, and a research flow chart has been added to the methodology section. The conclusion has been revised to motivate the academic community toward actionable, engaged scholarship.

Your critical assessment has been instrumental in these revisions, and we extend our deepest gratitude.

Claver Giovanni da Silveira Pinheiro

Round 2

Reviewer 2 Report

The authors addressed all changes very throughly. Very well done. 

Reviewer 3 Report

The authors did an excellent job in revising the manuscript.